# Challenges in the Medical and Psychosocial Care of the Paediatric Refugee—A Systematic Review

**DOI:** 10.3390/ijerph191710656

**Published:** 2022-08-26

**Authors:** Jakub Klas, Aleksandra Grzywacz, Katarzyna Kulszo, Arkadiusz Grunwald, Natalia Kluz, Mikołaj Makaryczew, Marzena Samardakiewicz

**Affiliations:** Department of Psychology, Medical University of Lublin, 20-093 Lublin, Poland

**Keywords:** paediatric refugees, refugee crisis, Ukraine

## Abstract

Background: After the invasion of Ukraine, neighbouring countries were forced to find systemic solutions to provide medical care to those fleeing the war, including children, as soon as possible. In order to do this, it is necessary to know the communication problems with refugee minors and find proposals for their solutions. Methods: A systematic review of the literature from 2016 to 2022 was conducted according to PRISMA criteria. Results: Linguistic diversity and lack of professional readiness of teachers are the main constraints hindering the assistance of refugee children in schools. Problems during hospitalization include lack of continuity of medical care and lack of retained medical records. Solutions include the use of the 3C model (Communication, Continuity of care, Confidence) and the concept of a group psychological support program. Conclusions: In order to provide effective assistance to refugee minors, it is necessary to create a multidisciplinary system of care. It is hoped that the lessons learned from previous experiences will provide a resource to help refugee host countries prepare for a situation in which they are forced to provide emergency assistance to children fleeing war.

## 1. Introduction

### 1.1. An Outline of the Escape of Refugees after the Russian Invasion of Ukraine

After weeks of tension and an escalation of the conflict in eastern Ukraine that began in 2014, Russian troops invaded Ukrainian territory on 24 February 2022. The escalation of the conflict has transformed the country’s already unstable situation into a full-scale state of emergency [1]. Major military attacks have been reported across Ukraine, including in the capital, Kyiv, and the Donetsk and Luhansk regions, among others. The situation remains extremely dangerous for everyone living in Ukraine, and the number of people forced to flee is rapidly increasing [2]. According to the 1 May 2022 update, it is estimated that more than 5.5 million refugees have fled to neighbouring countries since 24 February, and there is a continuing upward trend in this phenomenon. Additionally, more than 7 million Ukrainians are internally displaced within their country [3]. The majority of refugees are women and children (the latter group accounting for 40% of the refugee population), the majority of adult men crossing the border appear to be from outside Ukraine due to the ban on men with Ukrainian citizenship leaving the country. There are also reports of vulnerable and marginalized refugee groups, including the elderly, people with disabilities and ethnic minorities [4]. In the face of the refugee crisis resulting from the aforementioned conflict, several countries have had to confront the refugee crisis for the first time, including Poland, Romania, Moldova and Slovakia (Figure 1) [3]. Particular attention should be paid in the case of minors, for whom the situation of fleeing a country is often the first moment of losing a sense of security in their lives.

### 1.2. General Information on the Role of Communication in Medicine

The role of communication in medicine was studied and described back in the 20th century. The results of a 1995 systematic review showed a correlation between effective doctor–patient communication and improved patient health outcomes [5]. Later studies confirmed that both trust and communication in medicine were positively associated with patient satisfaction and perceived quality of healthcare services in terms of better adherence to medical recommendations, leading to better therapeutic outcomes of treatment and better perceived quality of healthcare services [6]. Practitioners who work to improve the way they express empathy and communicate positive messages are likely to result in improvements in the mental and physical state of many patients and improve their overall satisfaction with their care [7].

At the same time, studies have outlined the effects of poor communication. It has been found that poor communication can lead to a variety of negative consequences: lack of continuity of care, compromised patient safety, patient dissatisfaction and inefficient use of valuable resources, both in unnecessary research and physician time and economic consequences [8]. For example, communication difficulties played a role in the vast majority of medical accidents experienced by participants in a 2004 study [9]. Furthermore, the problem of communication may relate to the provision of continuity of medical care. Nurses participating in one study noted many deficiencies in transitions from hospital to specialised care facilities required repeated telephone explanations, caused delays in care (including delays in pain control), increased staff stress, frustrated individuals and family members, and directly contributed to a negative image of the specialised care facility and increased risk of hospital readmission [10]. In turn, inadequate coordination between institutions involved in assisting victims in an extreme situation, including health care, results in loss of resources, increased expenditure on assistance and often poor quality of healthcare [11].

With reference to the benefits of proper communication in medicine and the negative consequences in the case of poor and uncoordinated communication, systemic solutions should be sought to simultaneously provide medical assistance to fleeing refugees while preventing a reduction in the quality of the services provided to citizens of countries receiving refugees. With the sudden influx of such a large number of people in such a short space of time, it is essential to develop a management strategy to help as many people as possible in the most effective way. Particular care should be given to fleeing children, for whom the war situation will leave long-term consequences in terms of physical and mental health [12]. By having countries that have had to deal with the refugee crisis in the 21st century, their experiences should be used to outline the most important pillars for helping refugee minors.

In order to achieve good communication in medical facilities during the influx of paediatric refugees into the host country, it is additionally necessary to learn about the differences in the characteristics of the psychological profile between refugee minors and their peers from a war-affected country, and to find out what challenges the health system of the host country faces and what role non-medical facilities close to the refugee can play in assisting medical facilities in their activities. We believe that the knowledge gained will allow institutions to prepare in advance to effectively assist paediatric refugees, as they will know in advance what health and psychological problems underage refugees may present with, and will allow the better coordination of healthcare activities between the institutions themselves, increasing the quality of medical services and reducing hospitalisation times.

## 2. Materials and Methods

The aim of this paper is to report on the topic of communication between refugee minors and health care workers and other public actors from countries experiencing a refugee crisis. A systematic review of the scientific literature from 2016 to 2022 was conducted using PubMed and ResearchGate search engines, followed by a selection according to PRISMA-S Checklist criteria [13].

The thematic search involved finding scientific articles in the PubMed database from the years 2016 to 2022 (a period of increased number of papers emerging on paediatric refugee patients) meeting the criteria: [health care] AND [paediatric refugee], for ResearchGate: [paediatric] AND [refugee] AND [health care] in order to find articles meeting the article’s objectives as accurately as possible. In addition, 3 articles meeting the topic of the paper from other sources (Google Scholar) were included in the review. The search was restricted to English-language articles. Duplicate articles were removed and the abstracts of the received articles were reviewed. The co-authors selected papers containing relevant data on one of the three topics addressed in the paper: issues and characteristics of the health and psychological profile of refugee minors, difficulties related to the refugee crisis in different countries of the world, and proposals for systemic solutions to prepare a country at risk of a sudden influx of refugee minors. The article proposal lists were presented to the other co-authors dealing with the relevant subsections of the thesis, who decided to include a number of articles finally selected for review of the full versions of the articles contributing relevant content to the thesis. Detailed numerical information is provided in Table 1 and Figure 2.

## 3. Results

### 3.1. Characteristics of the Health and Psychosocial Situation of the Refugee Minor

The general health status of refugee children is worse than that of non-immigrant children. Infectious diseases such as tuberculosis, measles, malaria, hepatitis B, HIV and intestinal parasites are considered to be among the more serious health problems. An increased prevalence of syphilis and polio virus has also been reported [14,15]. In the study group, 33% of which were Syrian refugees, followed by Afghan and Egyptian refugees, it was found that only 35.8% of children were fully vaccinated against diphtheria, tetanus, pertussis and MMR. Vaccination rates against hepatitis B, pneumococcal, polio, chickenpox and HIB were much lower, not exceeding 17% [16].

A positive correlation was observed between the incidence of tuberculosis and the percentage of immigrants living in the region [17].

Infections are the most common cause of hospitalisation of refugee minors, and among children, respiratory infections are the most common (Syrian refugees) [18,19]. Studies on children permanently migrating to European countries have shown low vaccination rates against hepatitis B, chickenpox, measles, mumps, rubella and low immunity to tetanus and diphtheria. Additionally, many newborns born on the move may not have access to the screening for birth defects that is widely offered in European countries (mainly refugees from Afghanistan, Jordan, Lebanon, Syria and Turkey) [20].

The high proportion of children with severe nutritional problems is also a significant problem. Lack of access to sufficient, safe and nutritious food results in a low intake of micronutrients, fibre, fruit and vegetables and a higher incidence of both overweight and obesity and malnutrition resulting in developmental disorders (refugees from Myanmar, Afghanistan, Iraq and Iran) [21,22].

Refugee children are at high risk of toxic stress, i.e., extreme, frequent and persistent adverse events without the presence of a supportive carer. These events include, but are not limited to, the death of a family member, a life-threatening illness, a natural or man-made disaster or a terrorist incident. There is often social isolation, discrimination and isolation from family members, which greatly affects the child’s mental health. The prevalence of mental illness among this group is high. Post-traumatic stress disorder, depression and anxiety disorders are the most common [23,24]; 25% of refugee respondents (mainly Somalis, followed by Eritreans, Afghans and Syrians) experienced symptoms of PTSD [25].

Post-traumatic stress was also observed in 33% of Syrian children living in a refugee camp in Germany [26] (as of 2015). The average prevalence of anxiety disorders and major depression in conflict-affected populations is estimated to be two to four times higher than the estimated global prevalence. Minor refugees affected by armed conflict are particularly vulnerable to domestic violence, sexual violence and the breakdown of family structures [27]. They may also exhibit a wide range of stress reactions including specific anxiety, prolonged crying, lack of interest in their surroundings, psychosomatic symptoms and aggressive behaviour [28].

In a refugee camp in Turkey, the rate of depression was 60%, and symptoms such as aggressive behaviour also appeared (22%). Similar disorders are observed in Bangladesh, which received Rohingya fleeing the genocide in 2017. Of the 342 children assessed, 97% were hyperactive and 85% had behavioural problems. Half of the respondents were unable to build relationships with their peers [29].

Children may also experience acculturative stress related to the difficulties that may arise in maintaining their own traditions in a foreign environment and religious customs including, but not limited to, preparation for prayer (Afghan refugees) [30].

In studies, refugee minors also indicated negative perceptions of mental illness, the stigma that goes with it, and fear of the social consequences of having such an illness [31].

Among newly arrived refugees, almost 30% of patients were diagnosed with two or more conditions, demonstrating the complex diversity of conditions. The important role of family physicians in the ongoing care, as well as the comprehensive assessment of the health status of immigrants upon arrival in a new country (mainly refugees from Burma, Sudan, Afghanistan, Democratic Republic of Congo, and Thailand) is highlighted [32]. The quality of life of refugee children is largely affected by the high number of uncorrected visual impairments, which in the long term may affect their development and success in school. Parents are less likely to have their children’s eyesight checked, through limited information about the medical care available to them, as well as limited access to expert eye care and literature on eye problems (Syrian refugees, Burmese refugees) [30,33].

Inadequate oral hygiene and resulting tooth decay is also a serious problem among refugee and immigrant children. The main factors influencing its development are cultural differences and lack of knowledge of protective factors, but also financial constraints. Language and dental experience in the home country are also barriers [14,34]. The diagnosis of diabetes and its treatment among paediatric refugees has also become a challenge for Europe. Compared to the native population, refugees have been reported to have higher rates of severe hypoglycaemia, poorer diabetes control, and a higher incidence of long-term complications: microalbuminuria and diabetic retinopathy (mainly refugees from Africa, Syria, Iraq and Afghanistan) [35].

In addition to the aforementioned health problems, one of the reasons for the need for urgent medical care is unintentional injuries. Refugee immigrants have a 20% higher rate of unintentional injuries compared to non-refugee immigrants [36]. Surgical care in refugee children also includes routine surgical problems such as inguinal hernia and appendicitis, non-war-related injuries and gynaecological problems [16].

Among paediatric refugees, there is also a group of patients suffering from rare genetic diseases including Noonan syndrome, Laron syndrome, mitochondriopathy, Turner syndrome and arthrogryposis. They constitute a small but equally important group requiring regular medical consultations (Syrian, Eritrean, Afghan, Armenian, Somali, Algerian and Russian refugees) [37].

Another aspect is skin diseases, which are common among displaced populations. There may also be few or largely limited access to dermatologists in the host country, due to the distance of specialists from the refugee camp and thus the cost of transport [38]. Within weeks or even months of delays from medical diagnosis, skin infections may progress to a generalised form or cause an epidemic among the local community (African refugees) [39].

The specialised care of refugee patients is quite challenging due to the communication and language barrier, but also the lack of previous medical records, reports of past illnesses and surgical procedures (Syrian refugees) [40]. Another emerging difficulty is the many moves that often prevent continuity and accurate planning of further treatment [41]. Other stressors that inhibit the health-seeking behaviour of refugees are unfavourable housing and neighbourhood conditions, the challenges of interpreter services, the difficulty of navigating medical facilities, and the transportation to them themselves. These factors, as well as parents’ increasing depression and anxiety, prevent them from providing adequate medical care for their own children (Syrian refugees) [42].

### 3.2. Experiences from Recent Refugee Crises

Almost 85% of the world’s refugee population has been hosted in low- and middle-income developing countries. These countries, Lebanon, Jordan and Bangladesh among others, are unable to provide adequate living conditions for such a large number of admitted refugees and face a heavy burden on their health systems [43].

Data from Lebanon indicate that Syrian refugee children are very often injured while living in refugee camps. The most common type of injury was burns, which have been proven to be clearly due to the low socio-economic status of Syrian refugee families and poor housing conditions (e.g., flammable tents, heating and cooking on open fires, overcrowding). In addition, refugees have limited access to health care, lack insurance and the ability to cover medical costs. In Lebanon, there is a lack of systemic interventions to improve this situation, in contrast to numerous high-income countries where significant funding has been allocated for prevention, burns treatment and aftercare [44].

Statistics show that around 83% of refugee families in Lebanon have at least one person who suffers from a chronic illness, putting a significant strain on Lebanon’s already underfunded health system. One study analysed data on Syrian refugee children with heart defects. It noted that there is an increased rate of severe CHD among refugee children due to lack of access to health care, with only children with severe symptoms being able to receive medical attention. In developing countries, fewer patients reach adulthood due to delayed diagnosis, and inability to start or complete treatment. All costs associated with this examination and treatment of Syrian children with heart defects have been covered by non-governmental organisations (UNHCR, Brave Heart Fund, and GOLI), and medical professionals have often waived their salaries [45].

Another form of assistance to refugees is NGO-organised medical missions in Jordan such as the short-term SAMS medical mission in a camp for underage Syrian refugees. Infectious diseases and GDO infections were most common among the children treated. This was facilitated by the living conditions in the camp, including overcrowding, lack of access to drinking water and hygiene products. Typically, infections were severe due to delayed diagnosis and difficult treatment conditions—doctors received patients in tents without specialised equipment [46].

One study in Bangladesh highlighted the widespread lack of palliative care for refugees in humanitarian relief, an essential component during a humanitarian crisis. Many patients are experiencing pain and suffering due to severe injuries and illnesses, yet national regulations limit access to powerful painkillers. Neither the government nor NGOs are showing action to improve this situation. Among the refugees surveyed, as many as 62% (n = 96) experienced significant pain (62%, n = 96) and in 70% (n = 58), the prescribed treatments were ineffective (70%, n = 58). A total of 39.1% reported a need for medication and only 52.5% (n = 32) among them received their medication; 52.6% reported needing medical equipment and 72.0% (n = 59) of these patients did not have access to it [47].

In contrast to developing countries, highly developed countries face, usually, quite different problems regarding refugees. As a rule, they are provided with decent living conditions and care, but systemic solutions that are not fully effective are a problem.

According to the 2016 Australian guidelines, every arriving refugee must undergo a comprehensive health assessment. A medical examination is mandatory for all refugees and asylum seekers and should be conducted within 1 month of arrival in Australia. TB and Hepatitis B screening and booster vaccinations are also part of this procedure. These comprehensive screenings are carried out within the primary health care setting and with the assistance of nurses from the Special Refugee Health Program, who are also tasked with facilitating access to health care for newly arrived refugees. Refugees with confirmed active TB are treated prior to travel, and those with early or latent TB are referred to TB centres for follow-up in separate health facilities [48]. However, any asylum seeker must initially be placed in a special guarded facility and after an unspecified period of time be transferred to community custody. This process delays diagnosis and treatment (the median duration of detention was 7 months); moreover, multiple relocations do not allow for continuity of medical care. During detention, refugees do not have access to medical, psychological and educational services. An important case demonstrating the living conditions in community detention is the result of a 2014 Australian Human Rights Commission investigation. It showed that in detention, the majority of children experience physical and sexual assaults and there are numerous cases of self-harm and suicide [49].

Some inconsistency in health care for asylum seekers has been shown in Finland. The Finnish health care system for refugees is separate from general health care and funded by the immigration service. However, asylum seekers under the age of 18 have the same access to health services as Finnish children, including school health care. Despite this, the municipal authorities managing the medical facilities in their area have not provided services to refugees, have not been involved in combating infectious disease outbreaks and have denied public health services to refugee children. As a consequence, immigration services were forced to organise and pay for medical services in private facilities, where they were often incomplete or of poor quality. Even the introduction of further guidelines by the Finnish government did not influence the decisions of municipal authorities [50].

An important aspect concerning the lives of refugees in highly developed countries is marginalisation, discrimination and racism. In European countries, i.e., Italy, Switzerland and the Netherlands, the risk of immigrants being assaulted was much higher than in Canada or the USA [51].

Using Spain as an example, it can be concluded that unstable living situations and frequent relocation, limit access to healthcare and prevent continuity of treatment. In the study conducted, approximately 83% of the immigrants surveyed completed the entire treatment process, while only 22% completed the hepatitis B vaccination schedule. It should be noted that the medical care and vaccination offered were free of charge, and the failure of the treatment process was most likely due to the unfavourable factors mentioned earlier [52].

### 3.3. Systemic Solutions for Refugee Care

#### Adapting the Health Care System

Migrants and refugees have specific health needs while facing a number of barriers to accessing the health care system. Research indicates that language and cultural barriers have a negative impact on the healthcare of people with limited knowledge of the language spoken in the country, resulting in limited access, higher hospital admissions, increased risk of permanent damage and limited health knowledge due to communication difficulties [53]. In this context, it is important that medical professionals are adapted to work with migrants and refugees. A working model based on the 3C Model is useful for this purpose (Figure 3) [54]. Of the three, proper communication is identified as the most important. It is required to understand the patient’s problem and allows for the exchange of information about symptoms, presumed diagnosis, required diagnostic tests, treatment and prognosis [55]. In a pan-European survey of paediatric emergency workers, 60% indicated that language and translation problems are one of the most important barriers to providing care to immigrant children [56]. A second important factor is the continuity of health care dependent on refugees being informed and educated about the health care system of the country they are in, the ease of access to health care system facilities, the inclusion of medical appointments in the refugees’ personal schedule and the cooperation of different institutions and providers [54]. Concepts such as ‘family doctor’ and ‘preventive health care’ may be incomprehensible to some people, so clarification of such terms is necessary to ensure continuity of health care [57]. As refugees may have limited access to transport, it is beneficial to integrate medical visits with other visits related to the asylum process or education [58]. The last component of the model is trust. There are two main components: the development of trust in someone or something and the ability to control the situation. It is important to remember that establishing a trusting relationship is a two-way process and requires mutual education. In order to be in control of healthcare decisions, it is necessary to understand the facts and apply one’s own health beliefs and priorities to decision making. The ability to feel self-efficacy and anticipate decisions has been described as particularly important in refugee mental health [54].

Interpreter services play a key role in the healthcare system. An interpreter is a communication specialist trained to translate everything that is said, maintain confidentiality, ensure transparency and point out cultural differences that impede communication [54]. The use of professional interpreters in person results in a shorter total emergency department time compared to the use of professional interpreters over the phone [59].

Refugees, especially children, are exposed to multiple mental health risks. Repeated trauma in turn puts the refugee child at high risk of post-traumatic stress disorder (PTSD), anxiety, somatisation disorder and depression [60]. To address mental health problems such as PTSD, depression, behavioural problems and anxiety among refugee minors, the effectiveness of a psychological support programme was tested in a group of 32 participants. Each session lasted 70–90 min, the students were divided 8–10 to two teachers. An improvement in trauma-related symptoms was noted. The greatest reduction in symptoms was observed on the anxiety and intrusive symptoms scale. No significant changes in the categories of peer problems, behaviour and hyperactivity (Table 2) [61].

### 3.4. Medical Innovations in Working with Refugees

Estimates of the prevalence of PTSD among adolescent refugees range from 11 to 75 percent, while depression ranges from 4 to 47% [62]. In order to provide help for a condition as severe as depression, it is important to diagnose it early. Advances in medical information technology (health-IT) are enabling the development of multifaceted interventions that include provider training, screening and notification, and clinical decision support, which may be more effective than any single intervention to improve mental health outcomes in primary care [63]. Modern technologies can be adapted and used for any patient group with limited native language skills, but it is worth ensuring that the application is compatible with the patient’s cultural code.

It is also beneficial to include the targeted community in the development of solutions. This was demonstrated by the example of developing, according to the community-based participatory research (CBPR) model, an innovative vaccine education technology using virtual reality. The project targeted Somali refugees. The community was involved through a series of discussions, interviews with Somali parents, workshops and the use of virtual reality techniques in teaching [64].

### 3.5. Integration of School-Age Children

Refugee children often have learning and developmental problems, but come for help too late, failing to take advantage of early intervention or school and social support available in developed countries. Risk factors for problems at school include the experience of trauma (before or after migration), interrupted learning, low teacher expectations of school achievement, financial problems in the household, while protective factors include knowledge of the language of instruction in the new country, adequate recognition of the child’s educational experience by the school team, appropriate assessments and expectations, parental support in the education process and a high level of parental education [65].

The Turkish General Directorate of Teacher Training and Education has launched a series of national in-service training workshops on inclusive education, specifically the education of Syrian refugee children, children with disabilities and children affected by violence, immigration or natural disasters. The introduction of visuals, games or stories for Syrian children with Turkish subtitles has been shown to assist their integration into their new school environment [66]. In order to avoid Syrian children being ignored by Turkish children, it has been shown that educational efforts should be focused on integrating children of both nationalities, and that the work of a school psychologist is effective in removing any prejudice in Turkish children [67].

The health status of children and adolescents is significantly influenced by school nurses, who are the first of the health professionals to encounter refugee children and adolescents. Therefore, they may have a particular role in supporting them during such a difficult time. A qualitative study conducted in 2019 among Swedish nurses found that they need to expand their skills in recognising the needs of people in crisis and after traumatic events, expanding their knowledge of the culture of the refugee child’s country of origin [66].

## 4. Discussion

The fluctuating situation and the increasing number of displaced persons pose humanitarian and logistical challenges to the health systems of refugee-hosting countries. For example, Ukrainian refugees may face barriers to healthcare upon arrival in both transit and destination countries.

This review highlights the importance of changing the mindset between a paediatric refugee patient and a patient who has not experienced fleeing war. The authors of the study repeatedly highlighted the situation where, together with the arriving patient, the local health system had to face de facto problems of the health system of the country from which the refugee came, if only in the context of immunization [14,15,16]. Contrary to appearances, problems resulting from the lack of primary prevention do not only concern the poorest countries such as Syria or Afghanistan [20]. In addition, in the case of Ukrainian refugees, as a result of the low vaccination rate of the population compared to richer countries, patients may arrive with diseases already marginally diagnosed, such as measles or polio, as there were outbreaks of these diseases in Ukraine before the Russian invasion (in 2019 and 2021, respectively) [68,69,70,71]. There is also a risk of more patients with more advanced and less prognosis chronic diseases. Therefore, in our opinion, it is extremely important to review epidemiological reports from refugee countries so as to be ready for the need to combat diseases that are less common in the native population or the need to treat various diseases that are much less promising.

The experience of countries that have had to deal with the need to take in refugees in the past decade shows that there is a large spectrum of problems in providing assistance to displaced persons depending on the wealth of the country. While elementary health care needs, i.e., prevention, burn and post-burn care, availability of doctors or palliative care, are not being met in the poorest countries [44,46,47], in rich countries, attention has been drawn to difficulties in maintaining continuity of health care, availability of interpreters for displaced persons or lack of retained medical records [40,49,52]. The examples given highlight the complexity of the problem and the need to develop effective systemic solutions to improve the effectiveness of assistance to minors in both the poorest and richest regions of the world.

In order to provide effective assistance, it is necessary both to ensure rapid communication within the ward and between hospital departments, which requires financial outlays for, among other things, hiring interpreters, and to create a system in which the patients’ records are available in other facilities in the country or region, e.g., the European Union [59]. It is worthwhile to use such simple practice models as the 3Cs, through which priorities and the basis for building good relationships with foreign-speaking patients can be established more quickly [44]. It is also worthwhile to improve the Turkish psychological support programme for refugees so that it also addresses the issues of peer problems and hyperactivity of refugee minors [61].

The important role of the school in the process of integrating students must not be forgotten. The involvement of medical professionals working there, e.g., school psychologists and nurses, should be used to break down barriers between children speaking different languages to each other [66,67]. In addition, the role of teachers should be emphasised, as frontline workers, who could refer children in need of psychological help to facilities and places specialised in therapies for traumatised patients (e.g., psychological offices). As a result, refugee minors would gain more trust in state institutions, including those dealing with health care, so that they would report their health complaints more quickly and obtain professional medical help more quickly, which would ultimately lead to a bottom-up improvement in the communication of the refugee minor with the health care professional. Combined with the top-down facilities in hospital wards mentioned in the paragraph above, it would become possible to create a three-tier system that maximises the aspects of effective and rapid health assistance to refugee minors (Figure 4) [41,54,60,61,67,72].

## 5. Conclusions

The problem raised calls for systemic changes that are necessary to effectively help refugees. Let us not forget that today’s young patients are a group of intelligent and valuable people who, in a few years or so, may repay the societies that receive them in an emergency situation with their work, for example, filling jobs in professions that are understaffed in the country that has duly taken them in. We believe that the effort made to improve communication in medicine will bring with it long-term positive effects in every country where minors are helped.

## Figures and Tables

**Figure 1 ijerph-19-10656-f001:**
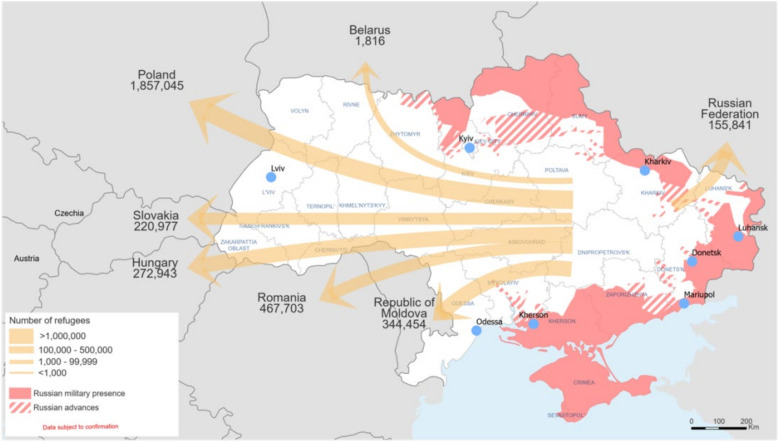
Polulation movement and displacement of refugees from Ukraine to neightbouring countries (as of 15 March 2022) (Reprinted from [3]).

**Figure 2 ijerph-19-10656-f002:**
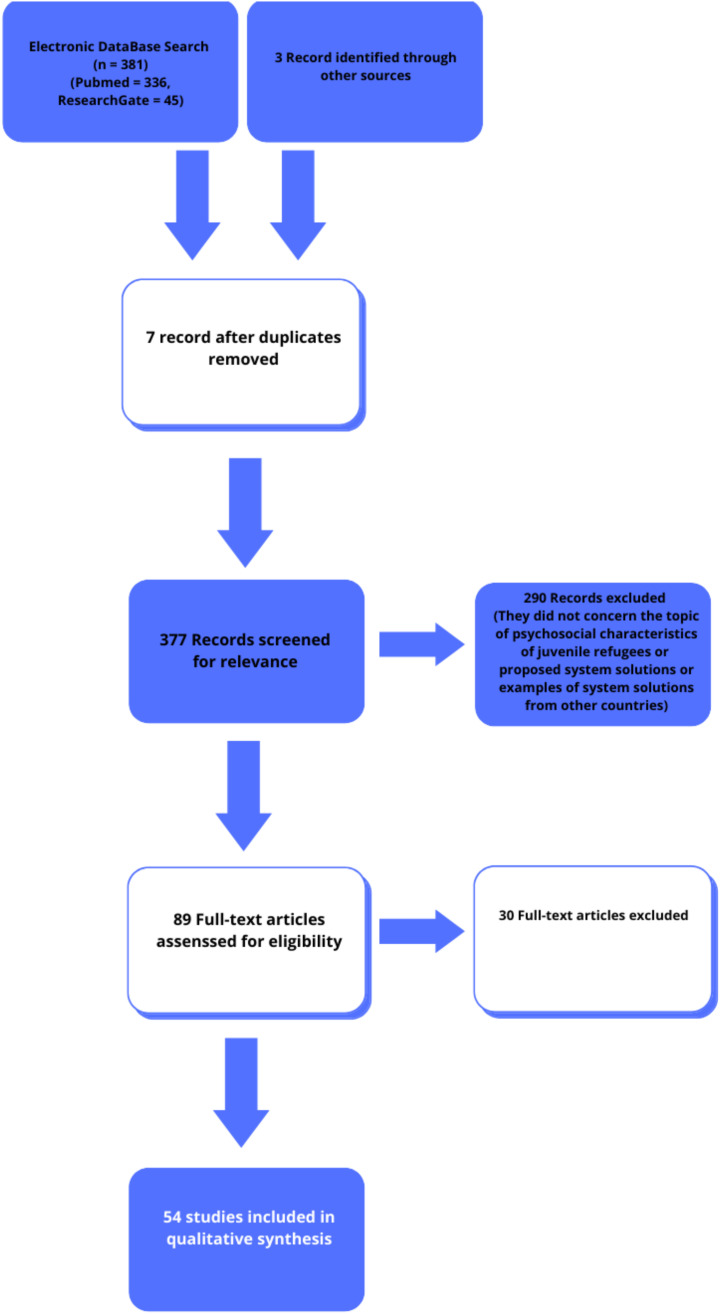
Figures from the systematic review (based on [13]).

**Figure 3 ijerph-19-10656-f003:**
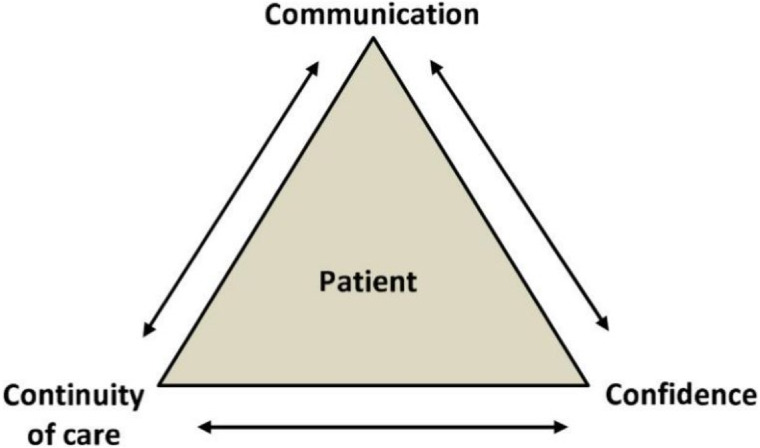
Diagram of model 3C [54].

**Figure 4 ijerph-19-10656-f004:**
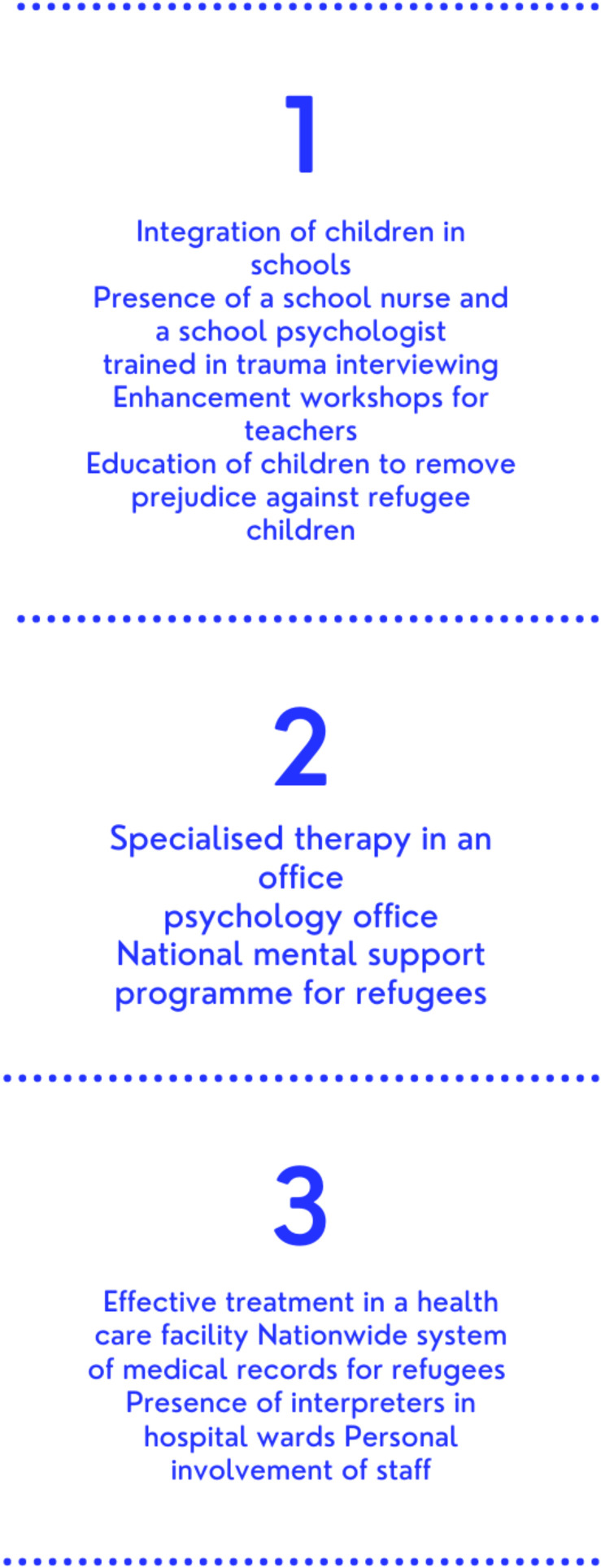
Proposal for a three-step system to improve communication between paediatric patients and medical facilities.

**Table 1 ijerph-19-10656-t001:** Summary of scientific articles used in the systematic review.

Authors, Publication Year	Number of Respondents/Group of Respondents	What Was Assessed/Country of Origin	Evaluation Methods	Main Result
Carrasco-Sanz A, Leiva-Gea I, Martin-Alvarez L, Del Torso S, van Esso D, Hadjipanayis A, Kadir A, Ruiz-Canela J, Perez-Gonzalez O, Grossman Z, 2018 [14]	n = 492, survey among European pediatricians	Evaluation of health care for migrant children. Survey coverage: member countries of the European Pediatric Research Network	Online survey	The overall health status of migrant children is worse than non-migrant children, with chronic diseases being the most common health problem. Cultural/language factors were reported as the most common barrier (90%) to accessing health care.
Kerbl R, Grois N, Popow C, Somekh E, Ehrich J., 2018 [15]	-	Evaluation of paediatric health care for refugee minors in Europe	Systematic review	An increased prevalence of infectious diseases in refugees and a significantly higher prevalence of mental disorders in refugees compared to residents have been described.
Loucas M, Loucas R, Muensterer OJ., 2018 [16]	n = 461, study of children aged 0.5–18 years	Surgical health needs of refugee minors/Syria, Afghanistan, Kosovo, Albania, Somalia, Eritrea, Serbia	Questionnaire	Previous surgical interventions were registered in 42.2% of participants. Among girls, genital mutilation was suffered by 11%. The most common mechanism of injury was a fall from a bicycle (38%), followed by burns (7.4%). As many as 20% of the children experienced physical violence during the flight or at their accommodation. Of the participants, only 63% were vaccinated as scheduled.
Fritschi N, Schmidt AJ, Hammer J, Ritz N, 2021 [17]	n = 139, survey of children aged 0–15 years	Assessment of childhood tuberculosis incidence and detailed diagnostic and treatment pathways/Eritrea, Somalia, Afghanistan, Brazil, Sudan	Assessment of childhood tuberculosis incidence and detailed diagnostic and treatment pathways/Eritrea, Somalia, Afghanistan, Brazil, Sudan	Among the 64 (46.0%) children born abroad, the incidence rates were higher. They reached a peak in 2016. The incidence rate was 13.7 per 100,000. The median interval between arrival in Switzerland and diagnosis of TB was 5 months, and 80% were diagnosed within 24 months of arrival. In 58% of cases, TB was confirmed by culture or molecular testing. Age >10 years, presence of fever or weight loss were independent factors associated with confirmed TB.
Yucel H, Akcaboy M, Oztek-Celebi FZ, Polat E, Sari E, Acoglu EA, Oguz MM, Kesici S, Senel, 2021 [18]	n = 728 survey of children aged 1 month to 18 years	The study aimed to analyse demographic data, clinical outcomes, treatment/management data and mortality data of hospitalised refugee children/Syria, Iraq, Afghanistan	Retrospective evaluation based on electronic medical records	Refugees hospitalised in the paediatric intensive care unit were significantly younger (median age 3.7 years).The most common cause of hospital admission was infection (51.09% of patients), often accompanied by other diseases. Mortality in the general paediatrics department was 16.4% for refugee patients and 8.6% for non-refugee patients. Factors associated with mortality were younger age and being a refugee.
Kampouras A, Tzikos G, Partsanakis E, Roukas K, Tsiamitros S, Deligeorgakis D, Chorafa E, Schoina M, Iosifidis E., 2019 [19]	n = 220, study of children <18 years	Assessment of morbidity and overall disease burden of refugee children/Afghanistan, Syria	Register of patients seeking medical advice	The majority of visits by children under 12 years of age were for infectious diseases (80.8%). The most common sites of infectious diseases among children were the respiratory tract (66.8%). Non-communicable diseases were mostly due to gastrointestinal disorders. Infants, toddlers and children were more likely to suffer from respiratory tract infections, while adolescents and adults were more likely to have non-communicable diseases.
Kadir A, Battersby A, Spencer N, Hjern A., 2019 [20]	-	The majority of visits by children under 12 years of age were for infectious diseases (80.8%). The most common sites of infectious diseases among children were the respiratory tract (66.8%). Non-communicable diseases were mostly due to gastrointestinal disorders. Infants, toddlers and children were more likely to suffer from respiratory tract infections, while adolescents and adults were more likely to have non-communicable diseases.	Systematic review	Low vaccination rates against hepatitis B, chickenpox, measles, mumps, rubella and low immunity against tetanus and diphtheria have been demonstrated. Additionally, many newborns born on the move may not have access to the screening for birth defects that is widely offered in European countries (mainly refugees from Afghanistan, Jordan, Lebanon, Syria and Turkey)
Kroening ALH, Dawson-Hahn E., 2019 [21]	-	Review article on health considerations of migrant and refugee children	Systematic review	Immigrant and refugee children are at increased risk of physical, emotional and behavioural health problems. Health issues affecting immigrant and refugee children should be framed within an ecological context that includes considerations of family, community and sociocultural influences. It is important to understand the migrant child’s migration history (or their family history), which or their family history), which provides a context for screening for infectious diseases and exposure risk (including trauma).
Newman K, O’Donovan K, Bear N, Robertson A, Mutch R, Cherian S., 2019 [22]	n = 1131, study of children aged 2 months to 17.8 years	Assessment of nutritional status and growth of paediatric refugees/Burma, Afghanistan, Iraq, Iran	Standardised dietary, medical and socio-demographic assessments of new refugee patients using a multidisciplinary paediatric refugee health service were analysed	Nutritional deficiencies were common, but varied by ethnic group: iron deficiency (12.3%), anaemia (7.3%) and inadequate dairy intake (41.0%). One third of the children did not consume meat. In infants under 12 months of age, breastfeeding was sustained (77.8%).
Harkensee C, Andrew R., 2021 [23]	n = 80 survey of children <18 years of age	Identify health needs and barriers to accessing health care/ Syria, Iraq, Nigeria, Albania, El Salvador, Libya, Somalia, Afghanistan, Namibia, Sudan	Mixed methods study (retrospective analysis of routinely collected service data, qualitative data from focus groups) of children attending a hospital-based specialist outpatient clinic	The most common diagnoses given are: anaemia, neurodevelopmental disorders, respiratory diseases. Mild to moderate stunting (23%), overweight and obesity (41%), stunting with obesity (9%) and micronutrient deficiencies (vitamin D (66%), vitamin A (40%) and visible (14%) or latent (25%) iron deficiency anaemia). 62% of children experienced psychological trauma and 39% had abnormal psychosocial wellbeing screening results. 21% of children required Level II or III care, 8% mental health referrals and 47% were observed at this specialist clinic. Unresolved health needs and significant barriers to accessing healthcare were reported.
Murray JS., 2018 [24]	-	The aim of this article was to describe the phenomenon of toxic stress and its impact on the physical and mental health of refugee children in Syria	Systematic review	It has been established that the prolonged brutal and traumatising war in Syria is having a profound impact on the physical and mental health of refugee children at an alarming rate. Preventing toxic stress should be a primary goal of all paediatric health professionals working with refugee children.
Kloning T, Nowotny T, Alberer M, Hoelscher M, Hoffmann A, Froeschl G., 2018 [25]	n = 154 survey of children aged 10–18	Investigate the morbidity profile and socio-demographic characteristics of unaccompanied refugee minors/Mainly Somalia, Eritrea, Afghanistan and Syria	Retrospective cross-sectional study including data from medical registries	Only 12.3% of all participants had no clinical symptoms on arrival. The main health symptoms were skin diseases and psychiatric disorders. Hepatitis A immunity was 92.8%, but only 34.5% showed a constellation of hepatitis B immunity. Suspected cases of tuberculosis were found in 5.8%. There were no HIV-infected individuals in the cohort. Two women were found to have undergone genital mutilation.
Yayan EH, Düken ME, Özdemir AA, Çelebioğlu A., 2020 [26]	n = 1115 survey of children aged 9–15	Investigating levels of post-traumatic stress, depression and anxiety in children living in refugee camps/Syria	Research data were collected using the Posttraumatic Stress Response Index, the State and Trait Anxiety Inventory for Children and the Children’s Depression Inventory	The results of the study showed that refugee children have physical and psychosocial health problems and experience high levels of post-traumatic stress, depression and anxiety. Most of the participating children (74%) were smokers. Anxiety and depression were statistically significantly related to post-traumatic stress.
Bendavid E, Boerma T, Akseer N, Langer A, Malembaka EB, Okiro EA, Wise PH, Heft-Neal S, Black RE, Bhutta ZA., 2021 [27]	-	Article on the impact of armed conflict on the health of women and children	Systematic review	The average prevalence of anxiety disorders and major depression in conflict-affected populations is estimated to be two to four times higher than the estimated global prevalence. Adolescent refugees affected by armed conflict are particularly vulnerable to domestic violence, sexual violence and the breakdown of family structures.
Bürgin D., 2022 [28]	-	Article on the impact of war and forced displacement on children’s mental health	Systematic review	The experiences that children have to endure during and as a result of war are in sharp contrast to their developmental needs and right to grow up in a physically and emotionally safe and predictable environment. Mental health and psychosocial interventions for children affected by war should be multi-level, child-centred, trauma-informed and strength and resilience oriented.
Khan NZ, Shilpi AB, Sultana R, Sarker S, Razia S, Roy B, Arif A, Ahmed MU, Saha SC, McConachie H., 2019 [29]	n = 622 survey of children <16 years of age	Investigating the level of neurodevelopmental and mental disorders in refugee children/Mjanma	Developmental Screening Questionnaire (DSQ; <2 years ) i Ten Questions Plus (TQP) andoraz Strengths and Difficulties Questionnaire (SDQ; 2–16 years ).	The number and percentage of children with positive screening results was 6 (4.8%) in children up to 2 years and 36 (7.3%) in children aged 2 to 16 years. 52% of the children were in the abnormal range for emotional symptoms in the SDQ and 25% for problems with peers. Lack of parents and loss of one or more family members in a recent crisis appeared to be significant risk factors.
Rosenberg J, Leung JK, Harris K, Abdullah A, Rohbar A, Brown C, Rosenthal MS., 2022 [30]	n = 19 parents	Characterise the experience of parenting, education and health care services/Afghanistan, Pakistan	Interview	The data shows that parents were relieved upon arrival in the US of their previous concerns about safety risks to their children. Many expressed a sense of isolation when they first arrived in the States. Some described a growing local support system. Parents perceived that it was difficult for their children to maintain religious rituals and described difficulties in maintaining traditions. All parents reported that, prior to coming to the US, their families only received health care in emergencies, which differed from the prevention system present in the US.
Bin Yameen TA, Abadeh A, Lichter M., 2019 [31]	n = 274 Survey of children <18 years of age	Assessment of eye care and visual impairment among refugee minors/Syria	Surveys, eye screening, eye examinations	The prevalence of uncorrected vision was 17.2 per cent for distance, 4.7 per cent for near and 0.7 per cent for distance and near vision, including loss of vision. Of these, 95.3% had not visited an ophthalmologist in the past year and 25.2% of parents were dissatisfied with their children’s vision. Presenting visual acuity in the better-sighted eye was 20/50 or less in 5.8%. This rate is 32 times higher than the prevalence rate in the average Canadian paediatric population (0.17%).
Masters PJ, Lanfranco PJ, Sneath E, Wade AJ, Huffam S, Pollard J, Standish J, McCloskey K, Athan E, O’Brien DP, Friedman ND., 2018 [32]	n = 291 refugees ≥16 years old—165 <16 years old—126	Review of reasons for referral, prevalence of conditions and outcomes for refugee patients/Mainly Burma, Sudan, Afghanistan, Democratic Republic of Congo	A retrospective review of patients attending the refugee health clinic at Geelong University Hospital	The most common diagnoses were latent tuberculosis infection (LTBI) (54.6%), vitamin deficiencies (15.8%), hepatitis B (11%) and schistosomiasis (11%). Less than two-thirds of patients completed LTBI treatment; 35.4% of patients attended all scheduled clinic visits.
Hussain AE, Al Azdi Z, Islam K, Kabir AE, Huque R., 2020 [33]	n = 670 neonatal and infant study 0–59 days old	Prevalence of eye problems among infants residing in refugee camps/Mjanma	Screening form, consent form and vision problems questionnaire	The most common problem among infants was tearing of the eye (14.8%). Visual inattention was identified as the second most common problem in infants reported by mothers (5.1%).Eye redness was observed in 4%. No eye problem was related to the gender of the infants.
Crespo E., 2019 [34]	-	Article on the importance of oral health in immigrant and refugee children	Systematic review	Poor oral hygiene and resulting tooth decay is also a serious problem among refugee and migrant children.
Prinz N, Konrad K, Brack C, Hahn E, Herbst A, Icks A, Grulich-Henn J, Jorch N, Kastendieck C, Mönkemöller K, Razum O, Steigleder-Schweiger C, Witsch M, Holl RW., 2019 [35]	n = 43,137 patients <21 years old n = 365—refugees born in the Middle East n = 175—refugees born in Africa n = 42,597—native-born patients in Germany/Austria	Diabetes care for paediatric refugees/Morocco, Egypt, Eritrea, Somalia, Ethiopia, Tunisia, Syria, Afghanistan, Iran, Iraq, Germany, Austria	Data were collected during routine care and taken from the standardised diabetes patient follow-up (DPV; “Diabetes-Patienten-Verlaufsdokumentation”—database for people with diabetes)	HbA1c and microalbuminuria were highest among refugees. African children were significantly more likely to experience severe hypoglycaemia. Hypoglycaemia with coma and rethionopathy were significantly more frequent in Middle Eastern children compared to the native population. Insulin pumps were used in a significantly higher proportion of native patients.
Saunders NR, Macpherson A, Guan J, Guttmann A., 2018 [36]	n = 999,951—total number of immigrants 153,822—refugees 846,129—non-refugees 0–24 years	Unintentional injuries among immigrant and refugee children/ The majority of immigrants were from South Asia and East Asia and the Pacific. The largest proportion of refugees came from South Asia and Africa	Cross-sectional population-based survey using linked health, administrative and immigration data	There were 6596.0 and 8122.3 emergency department visits per 100,000 non-refugee and refugee migrants, respectively. Hospitalisation rates were 144.9 and 185.2 per 100,000 in each of these groups. Unintentional injury rates among refugees were 20% higher than among non-refugees. j. Young age, male gender and high income were associated with injury risk. Compared to non-refugees, refugees had a higher rate of injury from most causes.
Buser S, Brandenberger J, Gmünder M, Pohl C, Ritz N., 2021 [37]	n = 19 children aged 0–16.7	Assessment of characteristics of asylum-seeking children with medical complexity detailing their underlying medical conditions and management/ Syria, Eritrea, Afghanistan, Somalia, Algeria, Russia	A retrospective cross-sectional study. Patients were identified by administrative electronic records	A total of 34/811(4%) visits were hospital admissions, 66/811 (8%) emergency department visits and 320/811 (39%) outpatient department visits. In children <2 years, genetic diseases and nutritional problems were the most common; in adolescents, orthopaedic diseases and mental health problems. Children with medical complexity seeking asylum represent a small but important group of patients requiring frequent medical consultations.
Knapp AP, Rehmus W, Chang AY., 2020 [38]	-	Review article on skin diseases in displaced populations	Systematic review	There may also be few or largely limited access to dermatologists in the host country, due to the distance of specialists from the refugee camp and thus the cost of transport, making it difficult to treat displaced persons effectively.
Kassem R, Shemesh Y, Nitzan O, Azrad M, Peretz A., 2021 [39]	n = 76 children aged 0–8 years	Determination of clinical features and response to treatment of tinea capitis among refugee children/Eritrea	Analysis of electronic medical recordsj	The most common clinical sign was peeling skin. Cultures were positive in 64 (84%) and direct examination in 65 (85%) cases, with a positive correlation between methods in 75% of cases. The most common fungal strain was T. violaceum. Treatment with fluconazole failed in 27% of cases. Griseofulvin 50 mg/kg/day was administered to 74 (97%) children and induced a clinical response. No side effects were reported.
User IR, Ozokutan BH., 2019 [40]	n = 254 survey of children aged 0–16	Assessing the sociodemographic and medical characteristics of childhood refugee patients and identifying their health problems/Syria	A retrospective cross-sectional study. Data were based on the records of patients admitted to the paediatric surgery department of a teaching hospital	The most common diagnosis was inguinal urothelial pathology (n = 50, 19.7%), followed by foreign body ingestion (n = 37, 14.6%) and caustic oesophagitis (n = 22, 8.7%). In 24.4% of cases, the cause of admission was preventable trauma. Comorbidities were present in 49 patients (19.3%). Anaemia was detected in 23.2% of cases. Difficulties in communication, lack of previous medical history and advanced disease symptoms were challenges faced by caregivers.
Baauw A, Rosiek S, Slattery B, Chinapaw M, van Hensbroek MB, van Goudoever JB, Kist-van Holthe J., 2018 [41]	n = 68—number of reported cases (barriers to health care)	Gain insight into pediatricians’ perceived barriers to health care for refugee children by analyzing logistical issues reported by pediatricians	Analysis of reports obtained through the Dutch Paediatric Surveillance Unit	Paediatricians reported 68 cases of barriers to care, ranging from mild to severe impact on the health of refugee children, reported between November 2015 and January 2017. The frequent transfer of children between refugee centres was mentioned in 28 reports of lack of continuity of care. Unknown medical history (21/68) and poor transfer of medical records resulting in poor communication between health professionals (17/68) contributed to barriers to good medical care for refugee children, as did poor health knowledge (17/68) and cultural differences (5/68).
Alwan RM, Schumacher DJ, Cicek-Okay S, Jernigan S, Beydoun A, Salem T, Vaughn LM., 2020 [42]	n = 18—Syrian refugee parents	Exploring refugee parents’ beliefs, perspectives and practices regarding their children’s health/Syria	Interview	The analysis identified the most salient themes: stressors excluded health-promoting behaviours, parents perceived barriers to health and showed dissatisfaction with the health care system. Stressors included poor housing and neighbourhood, reliving traumatic experiences, depression and anxiety, and social isolation. Dissatisfaction included emergency room waiting times, lack of tests and prescriptions. Health barriers included missed appointments and inadequate transport, translation services, health literacy and care coordination. Parents reported resilience through faith, seeking knowledge, using natural resources and utilising community resources.
Joury E, Meer R, Chedid JCA, Shibly O., 2021 [43]	n = 910, Syrian refugee children attending 5 primary schools for Syrian refugees in informal settlements in Bekaa, Lebanon	To assess the prevalence of oral diseases in the children of Syrian refugees living in Lebanon and to investigate their association with the duration of displacement	Cross-sectional study, analysis of data from a cross-sectional oral health needs assessment conducted by Global Miles for Smiles	The study highlights the importance of untreated tooth decay and pain in the refugee child population. Children with long-term resettlement had significantly more teeth with untreated caries compared to children who had been resettled for less than five years.
Al-Hajj S, Pike I, Oneissi A, Zheng A, Abu-Sittah G., 2019 [44]	n = 347—children of refugees settled in lebanon, age 0–19 years	Analysis of burn cases among Syrian refugee children in Lebanon	Retrospective cohort study	Data from Lebanon shows that Syrian refugee children are very often traumatised already living in refugee camps. In addition, refugees have limited access to healthcare, lacking insurance and the ability to cover medical costs. In Lebanon, there is a lack of systemic interventions to improve this situation, in contrast to numerous high-income countries, where significant funding has been allocated for prevention, burns treatment and aftercare.
Mostafa H, Rashed M, Azzo M, Tabbakh A, El Sedawi O, Hussein HB, Khalil A, Bulbul Z, Bitar F, El Rassi I, Arabi M., 2021 [45]	n = 439, Syrian refugees under 18 years of age, mean age 3.97 years	Describing the presentation, diagnoses, treatment, financial burden and outcomes among Syrian refugees with congenital heart disease (CHD) in Lebanon	A retrospective study, reviewing the medical records of all Syrian paediatric patients referred to the Children’s Heart Centre at the American University Medical Centre in Beirut for evaluation between 2012 and 2017	There is an increased rate of severe CHD among refugee children due to lack of access to health care, with only children with severe symptoms being able to receive medical attention. In developing countries, fewer patients reach adulthood due to delayed diagnosis, inability to start or complete treatment. All costs associated with this screening and treatment of Syrian children with heart defects have been covered by NGOs (UNHCR, Brave Heart Fund, GOLI), and medical workers have often waived their salaries.
Hamdan-Mansour AM, Abdel Razeq NM, AbdulHaq B, Arabiat D, Khalil AA., 2017 [46]	n = 250 Syrian refugee children aged 6–18 years	Investigating the physical and psychosocial health of displaced Syrian refugee children in Jordan	Cross-sectional exploratory design, structured questionnaires, data collection through face-to-face interviews	Evidence suggests that Syrian refugee children have been exposed to or witnessed serious traumatic war events. Exposure to these war events has been associated with negative psychological consequences for children, such as post-traumatic stress disorder (PTSD), depression and somatic disorders. Previous research has shown that children who survived the war continue to suffer from its psychological consequences for many years after the war. Large-scale interventions are therefore needed to address the mental health and physical trauma of refugees.
Doherty M, Power L, Petrova M, Gunn S, Powell R, Coghlan R, Grant L, Sutton B, Khan F., 2020 [47]	n = 311 people, including 156 people living with serious health problems and 155 carers	Characteristics of illness-related suffering and palliative care needs in refugees and caregivers settled in Rohingya refugee camps in Bangladesh	Cross-sectional study	Attention was drawn to the widespread lack of palliative care for refugees in humanitarian assistance, which is an essential component during a humanitarian crisis. Many patients are experiencing pain and suffering due to severe injuries and illnesses, yet national regulations limit access to strong pain medication. Neither the government nor NGOs are showing action to improve this situation. The physical, mental and social needs of patients and their caregivers are not being met by humanitarian aid. They do not have access to medicines and medical care.
Heenan RC, Volkman T, Stokes S, Tosif S, Graham H, Smith A, Tran D, Paxton G., 2019 [48]	n = 128, Syrian and Iraqi children aged 0–17 years receiving specialist migrant health care from January 2015 to September 2017 in Australia	To explore the health assessment of refugee children settled in Australia in the context of screening, the refugee primary health care model and the guidelines introduced	Retrospective audit of medical records of refugee children receiving RCH IHS services or outreach clinics	According to the 2016 Australian Guidelines, every arriving refugee should undergo a comprehensive health assessment. A medical examination is mandatory for all refugees and asylum seekers and should be conducted within 1 month of arrival in Australia. This also includes screening for TB and Hepatitis B and booster vaccinations. Despite good access to primary health care and screening, these activities are ineffective, not in line with current guidelines and not well supported by the RHP.
Hanes G, Chee J, Mutch R, Cherian S., 2019 [49]	n = 110, refugee children seeking asylum (<16 years), mean age 6 years	Determining the needs of paediatric patients among asylum seekers in Western Australia, the range of health and psychosocial problems they face and the associated challenges facing the Australian health system	Audit of multidisciplinary RHS assessments, medical records and hospital admissions of new asylum seeker patients (<16 years) between July 2012 and June 2016.	Australia’s system of reception and care for refugees, to some extent, hinders their diagnosis and treatment. The process delays diagnosis and treatment (the median duration of detention was 7 months), in addition, multiple relocations do not provide the opportunity to maintain continuity of medical care. During detention, refugees do not have access to medical, psychological and educational services (this is also affected by the visa category held). It has also been shown that in community detention most children experience physical and sexual assaults and there are numerous cases of self-harm and suicide.
Tuomisto K, Tiittala P, Keskimäki I, Helve O., 2019 [50]	-	Analysis of problems related to health care and access to medical services among asylum seekers in Finland	Qualitative review	The work identified three main problems in the governance of the asylum seeker health system: (1) Ineffective national coordination and governance; (2) Inadequate legal and supervisory frameworks leading to ineffective governance; (3) Disparities between constitutional health rights, statutory entitlements to benefits and available medical services.
Saunders NR, Guan J, Macpherson A, Lu H, Guttmann A., 2020 [51]	n = 22,969,443 (20,012,091 non-immigrants and 2,957,352 immigrants; 51.3% male and 48.7% female)	Exploring the association of immigrant or refugee status with experiences of violence and assault in Ontario, Canada	Population-based cohort study, used linked health and administrative databases	A Canadian study found that refugee youth in the country experienced assault as much as 51% less frequently than Canadian youth. The low rates of experiencing assault by migrants, including refugees, indicate a high level of support for immigrant settlement in Canada and suggest the influence of cultural factors on assault risk.
Serre-Delcor N, Ascaso C, Soriano-Arandes A, Collazos-Sanchez F, Treviño-Maruri B, Sulleiro E, Pou-Ciruelo D, Bocanegra-Garcia C, Molina-Romero I., 2018 [52]	n = 303, Median age was 28.0 years	Description of the health status of asylum seekers in Spain	A retrospective population-based study including all asylum seekers who applied for a medical examination at the Vall d’Hebron-Drassanes Tropical Medicine and International Health Unit (Barcelona, Spain) between July 2013 and June 2016.	Unstable living situations and frequent relocation, limit access to healthcare and prevent continuity of treatment. Language and cultural barriers, also hinder effective care. In the survey conducted, about 83% of the immigrants surveyed completed the entire treatment process, while only 22% completed the hepatitis B vaccination schedule. It should be emphasised that the medical care and vaccination offered was free of charge, and that the failure of the treatment process was most likely due to the previously mentioned unfavourable factors.
Clarke SK, Jaffe J, Mutch R., 2019 [53]	-	What communication barriers may be encountered when working with refugees? How can these barriers be overcome and what ethical challenges may be encountered when working with interpreters?	Przegląd systematyczny	Training by professional interpreters should be available at medical universities and hospitals, as well as remote access to interpreters. Collaborating as part of an interdisciplinary team can help avoid ethical dilemmas, help reduce disparities and ultimately save time and money.
Brandenberger J, Tylleskär T, Sontag K, Peterhans B, Ritz N., 2019 [54]	-	A summary of current knowledge on the provision of health care to migrants and refugees in high-income countries.	A systematic review of the literature from 2000–2017. Of the 185 articles found, 35 were selected for final analysis.	The 3C model provides a simple and comprehensive patient-centred summary of the key challenges in providing healthcare to refugees and migrants. It is supportive of health professionals, but the model itself is influenced by factors such as the specific regional context with legal, financial, geographical and cultural aspects.
Rahman A., 2016 [55]	-	Assessing the readiness of Canada’s health care system to accommodate Syrian refugees, Canada	Systematic review	The health system must adapt to meet the health needs of refugees by providing special training and mentoring for medical students and GPs on current health policies related to refugees, providing interpreters, and fostering collaboration between government and community organisations to develop outreach programmes.
Schrier L, Wyder C, Del Torso S, Stiris T, von Both U, Brandenberger J, Ritz N., 2019 [56]	-	To obtain consensus on recommendations for medical care for migrant children (asylum seekers and refugees). The work of authors from research centres in 7 countries	Recommendations; Current clinical guidelines and recommendations for the management of migrant children in the EU/EEA area were collected and compared	The document can serve as a tool to ensure basic rights so that migrant children in Europe receive comprehensive, patient-centred health care, access to interpreter services and specific recommendations for the prevention or early detection of communicable and non-communicable diseases.
Worabo HJ, Hsueh KH, Yakimo R, Worabo E, Burgess PA, Farberman SM., 2016 [57]	n = 39, (1) clients of an urban refugee resettlement agency in the Midwest in the last 5 years; (2) 18 years of age or older; (3) from Iraq, Eritrea, Bhutan or Somalia	The aim was to better understand newly resettled refugees’ perceptions of US healthcare. The data collected is expected to reduce health disparities and improve the quality of care, as well as provide direction for future solutions.", USA	4 in-depth interviews, analysis using the Colaizzi approach	Suggested strategies to improve the quality of health services include improving the cultural competence of health care providers, increasing access to English language classes and translation services, providing health materials translated into refugee languages, establishing stronger links between primary health care and resettlement agencies, and adopting innovative models of health care delivery.
Fazel M, Garcia J, Stein A., 2016 [58]	n = 40, uchodźcy w wieku nastoletnim	Access to needed mental health services can be particularly difficult for newly arrived refugee and asylum-seeking adolescents, although many are attending school, UK	This study explored young refugees’ impressions and experiences of mental health services integrated into the school system. Semi-structured interviews were conducted	Uncertainty in the asylum process has a negative impact especially on the social functioning and ability of young people to focus in school. Teachers play an important role in the support and integration process of refugee youth.
Boylen S, Cherian S, Gill FJ, Leslie GD, Wilson S., 2020 [59]	-	The aim of the review was to identify, critically appraise and synthesise the evidence on the impact of professional interpreters on hospital outcomes for migrant and refugee children with limited English proficiency, Australia	The aim of the review was to identify, critically appraise and synthesise the evidence on the impact of professional interpreters on hospital outcomes for migrant and refugee children with limited English proficiency, Australia	The use of ad hoc interpreters or the absence of an interpreter is inferior to the use of professional interpreters. The way in which interpreters are provided should be based on availability, accessibility, language requirements and patient preferences.
Hodes M, Vostanis P., 2019 [60]	-	Review integrates the latest research on risk and protective factors for psychopathology with service and treatment issues, UK	Systematic review	Many refugee children show resilience and function well, even in the face of severe adversity. The most robust findings for psychopathology are that PTSD and post-traumatic and depressive symptoms are more common in those who have been exposed to war experiences. Their severity may decrease over time with displacement, but PTSD in the most exposed may show higher continuity.
Gormez V, Kılıç HN, Orengul AC, Demir MN, Mert EB, Makhlouta B, Kınık K, Semerci B., 2017 [61]	32 participants aged between 10 and 15 years (mean = 12.41, SD = 1.68), mostly female (m/f = 12/20) were randomly selected from a sample of 113 refugee students based on their trauma-related psychopathology as reflected in the total Child Post-Traumatic Stress—Reaction Index score	Evaluation of an innovative group cognitive-behavioral therapy program delivered by trained teachers to reduce emotional distress and improve psychological functioning among war-affected Syrian refugee children living in Istanbul, Turkey	The effectiveness of the intervention was assessed by comparing pre-test/post-test using the CPTS-RI, Spence Children’s Anxiety Scale (SCAS) and Strengths and Difficulties Questionnaire (SDQ)	Protocol-based interventions need to be tested in controlled projects and larger samples so that a well-established intervention can be created and disseminated to provide psychosocial support to this vulnerable and traumatised population.
Green AE, Weinberger SJ, Harder VS, 2020 [62]	The sample included 301 patients aged 0–18 years; one infant was excluded and five 18-year-olds were retained for the final study sample of 300 patients aged 4–18 years	Study aims to conduct mental health screening of refugee children, United States	The Strengths and Difficulties Questionnaire (SDQ) was used as a screening tool for childhood mental health problems in primary care	The SDQ is a useful mental health screening tool in primary care for newly arrived refugee children. Doctors should screen on arrival to identify difficulties that may require intervention.
Sorkin DH, Rizzo S, Biegler K, Sim SE, Nicholas E, Chandler M, Ngo-Metzger Q, Paigne K, Nguyen DV, Mollica R., 2019 [63]	A randomised group of 18 primary care providers was allocated 390 Cambodian-American patients	A multi-component health information technology screening tool was developed to help providers identify and treat major depressive disorder and post-traumatic stress disorder (PTSD) in primary care settings, Canada	Group randomised controlled trial	This innovative approach provides an opportunity to train primary care providers to diagnose and treat trauma patients, most of whom seek psychiatric care in primary care.
Streuli S, Ibrahim N, Mohamed A, Sharma M, Esmailian M, Sezan I, Farrell C, Sawyer M, Meyer D, El-Maleh K, Thamman R, Marchetti A, Lincoln A, Courchesne E, Sahid A, Bhavnani SP., 2021 [64]	60 adult refugees from Somalia and seven experts who specialise in healthcare and autism research	Testing the effectiveness of an immunisation education platform using virtual reality (VR) to engage Somali refugee communities. United States	Community-based participatory research (CBPR) methods, including focus group discussions, interviews and surveys, were conducted with Somali community members and counsellors to design the educational content. A co-design approach with community input was used in a phased approach to develop the VR storyline	The CBPR approach can be used effectively to co-design an educational programme. Additionally, cultural and linguistic differences can be incorporated into an educational programme and are important factors for effective community engagement. Finally, effective use of VR requires flexibility so that it can be used among community members with different levels of health and technology awareness.
Minhas RS, Graham H, Jegathesan T, Huber J, Young E, Barozzino T., 2017 [65]	-	Designating the role of paediatricians and family physicians in caring for the developmental health of refugee children as a means of supporting their developmental and educational potential, Canada	-	The authors suggest using EMPOWER (Education, Migration, Parents and Family, Outlook, Words, Experience of Trauma and Resources), a mnemonic checklist they developed to assess developmental risk factors in refugee children. EMPOWER can be used in conjunction with online resources, such as Caring For Kids New to Canada, in providing evidence-based care for these children.
Musliu E, Vasic S, Clausson EK, Garmy P., 2019 [66]	n = 14, school nurses working with refugee children and young people	The aim of the study was to describe the experiences of school nurses in working with refugee children and adolescents	Semi-structured interviews were conducted with school nurses.	School nurses require development of tailored skills that focus on crisis, trauma and cultural awareness to meet the complex needs of working with refugee children and adolescents.
Karsli-Calamak E, Kilinc S., 2019 [67]	n = 5; three early childhood education teachers and two teachers of Turkish as a second language	The study aims to understand the changing experiences of teachers of Syrian refugee students in relation to inclusive education in Turkey	Field research in a public school located in a disadvantaged neighbourhood of the Turkish capital, where there were many Syrian refugee students. Three early childhood teachers and two teachers of Turkish as a second language were interviewed during the semestera	There should be critical conversations about the realities of refugee life resulting from structural obstacles. The process of constructing inclusive education requires prioritisation of all stakeholder claims and potential solutions to problems that directly affect them. There is a need for dialogue, transformative action and shared responsibility within and beyond national borders.

**Table 2 ijerph-19-10656-t002:** Class topics of the Turkish psychological support programme for refugees [61].

Session Numbers	Subject
1	Introductory activities to get to know each other and integrate into the group
2–3	Addressing strong emotions, maladaptive thinking styles, and creating alternative explanations for depressive, anxiety and stress-related experiences
4–6	Experiences of trauma and bereavement
7–8	Dealing with anxiety relapses, course summary

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
