# Peer review of "Challenges in the Medical and Psychosocial Care of the Paediatric Refugee—A Systematic Review"

_ijerph, 2022, doi:10.3390/ijerph191710656_

Round 1
Reviewer 1 Report
Dear Authors,
It is interesting topic, however there are some aspect , which should be corrected / explained in a better way:
1) The purpose of this paper is as following:
"The aim of this paper is to report on the topic of communication between refugee minors and health care workers and other public actors from countries experiencing a refugee crisis" (Materials and methods part)
it means that one of the main topic is communication, however:
in the last paragraph of point 1 (Introduction) the following research questions can be found:
"what characterizes a refugee minor compared to children who have not experienced fleeing their country? What are the challenges facing the health system of the refugee host country? What role can non-medical facilities close to the refugee play in assisting medical facilities with their activities?"
It arises a question: how these questions are related to the aim of the paper, mentioned in the beginning of point 2.
2) As from the aim of this article appears that the main clue is communication, therefore you should show it / discuss it in the introduction part. Introduction part is too poor - you provide only information on the directions and scale of refugees; -you should present arguments why you are going to focus on the communication.
Author Response
Dear Reviewer,
Thank you for taking the time to review our research paper. The review you have received contains a lot of valuable suggestions to make ours complete and valuable.
In this message we would like to address the comments received in the review:
1) Regarding question 1, we would like to inform you that the questions presented in the introduction served to expand the research problem stated in the abstract. The term "communication" is a broad concept, so in our opinion it was necessary to concretise which aspects of communication in medicine would be addressed in our research work. As a result of the feedback we received, we corrected the last paragraph of the introduction, changing the questions into indicative sentences, further clarifying which elements of communication need to be prioritised in order to effectively and efficiently help refugees fleeing to a non-war country (therefore, it is necessary to know the problems faced by refugee minors, what challenges medical facilities face in a situation of mass refugee flight, and the importance of coordination between medical facilities and other institutions that can play a significant role in maintaining the physical and mental health of children fleeing war).
2) Following the comment that the introduction of the paper was poorly presented, we have expanded this section to include information on the importance of communication in medicine. We believe that the included paragraph highlights the consequences of using both good and poor communication in medical settings. We would like to thank you again for this feedback, as with the addition of information on the importance of communication in medicine, it became possible to meaningfully clarify in the following paragraph the information about which there were doubts in point 1 of the feedback received.
A revised version of the manuscript is attached. In addition to the aforementioned corrections, the acknowledgements have been completed and cosmetic changes have been made regarding the numbering of references and bibliography, and we are therefore resubmitting an expanded version of the manuscript with the table showing the results of the systematic review so that the relevant reference number refers to the corresponding number in the table with the results of the systematic review. In accordance with the instructions received, the paper has been corrected in MS Word using the 'Truck Changes' function.
We kindly ask for positive consideration of the revised version of the scientific paper. We sincerely hope that the paper will be published in your prestigious journal.
With kind regards
On behalf of the research team
Jakub Klas
Department of Psychology
Medical University of Lublin
Reviewer 2 Report
The research is well done and comprehensive. The review of previous research covers the medical conditions that children present, as well as the psychological challenges they face as a result of exposure to highly traumatic events. The review also notes that refugee children are also exposed to marginalization and discrimination. They also face the challenges of language and cultural barriers in interfacing with refugee services and other aspects of the host country. In addition, their education is interrupted.
The model proposed straightforwardly addresses a way of handling all of these concerns in an orderly fashion. The situation of child refugees in Europe is, of course, much different than those in middle and low income nations where refugee children may face continuing risk and adversity in camps and centers. The authors might speak to how the model might work in those settings.
Author Response
Dear Reviewer,
On behalf of the research team, I would like to express my thanks for giving a positive review of our scientific work. After reading the review, we are extremely proud that the work was appreciated and judged to be of value.
With best regards
Jakub Klas
Round 2
Reviewer 1 Report
Dear Author,
There is an improvement and comments were taken into account.
Now, it is clear.